# Solution structure and pressure response of thioredoxin-1 of *Plasmodium falciparum*

**Claudia Elisabeth Munte, Hans Robert Kalbitzer***

Institute of Biophysics and Physical Biochemistry and Centre of Magnetic Resonance in Chemistry and Biomedicine, University of Regensburg, Regensburg, Germany

* hans-robert.kalbitzer@biologie.uni-regensburg.de

**Data Availability Statement:** The chemical shift values of NMR spectroscopy have been deposited in BioMagResBank database (BMRB ID 6442 and 16147) and the coordinates of the best NMR-

## Abstract

We present here the solution structures of the protein thioredoxin-1 from *Plasmodium falciparum* (*Pf*Trx-1), in its reduced and oxidized forms. They were determined by high-resolution NMR spectroscopy at 293 K on uniformly $^{13}$C-, $^{15}$N-enriched, matched samples allowing to identification of even small structural differences. *Pf*Trx-1 shows an α/β-fold with a mixed five-stranded β-sheet that is sandwiched between 4 helices in a $\beta_1$ $\alpha_1$ $\beta_2$ $\alpha_2$ $\beta_3$ $\alpha_3$ $\beta_4$ $\beta_5$ $\alpha_4$ topology. The redox process of the CGPC motif leads to significant structural changes accompanied by larger chemical shift changes from residue Phe25 to Ile36, Thr70 to Thr74, and Leu88 to Asn91. By high-field high-pressure NMR spectroscopy, rare conformational states can be identified that potentially are functionally important and can be used for targeted drug development. We performed these experiments in the pressure range from 0.1 MPa to 200 MPa. The mean combined, random-coil corrected $B_1^*$ values of reduced and oxidized thioredoxin are quite similar with -0.145 and -0.114 ppm GPa$^{-1}$, respectively. The mean combined, random-coil corrected $B_2^*$ values in the reduced and oxidized form are 0.179 and 0.119 ppm GPa$^{-2}$, respectively. The mean ratios of the pressure coefficients $B_2/B_1$ are -0.484 and -0.831 GPa$^{-1}$ in the reduced and oxidized form respectively. They differ at some points in the structure after the formation of the disulfide bond between C30 and C33. The thermodynamical description of the pressure dependence of chemical shifts requires the assumption of at least three coexisting conformational states of *Pf*Trx-1. These three conformational states were identified in the reduced as well as in the oxidized form of the protein, therefore, they represent sub-states of the two main oxidation states of *Pf*Trx-1.

## Introduction

Malaria is a disease typical of tropical and subtropical regions, caused by *Plasmodium* parasites. According to WHO, in 2021 [1] an estimated 247 million cases occurred in the 84 countries worldwide where malaria is endemic and around 619,000 people died. The disease mainly affects Africa (95% of all malaria cases; 96% of deaths) and more severely children, pregnant women, and patients with low immunity.

Of the 5 species that cause malaria in humans, *Plasmodium falciparum* is the most fatal. This parasite is becoming increasingly resistant to the hitherto most important drugs (sulphadoxine/pyrimethamine and chloroquine). Even for treatment with Artemisinin-based

derived structures were deposited in RCSB Protein Data Bank (PDB ID2MMN and 2MMO).

**Funding:** Deutsche Forschungsgemeinschaft (KA647/22-1) Funding of HRK The funders had no role in study design, data collection and analysis, decision to publish, or preparation of the manuscript.

**Competing interests:** The authors have declared that no competing interests exist.

Combination Therapy (ACT), currently recommended by the WHO, several cases of parasite resistance have already been reported (for a review see [2]).

Because the parasite spends part of its life cycle inside human erythrocytes, where it is exposed to very high oxidative stress, mechanisms that protect the parasite from this stress are of paramount importance for the survival of the parasite. The *Plasmodium falciparum* thioredoxin (*Pf*Trx) and the later discovered plasmoredoxin (*Pf*Plrx), which probably only occurs in *Plasmodium*, represent important regulatory redox systems and thus also important target molecules for the search for antimalarial drugs [3–5]. Furthermore, a glutathione system is found in *P. falciparum*, which includes glutathione reductase (GR), glutathione, glutathione-S-transferase, and glutaredoxin (Grx) [6, 7]. Inhibiting the redox system of the host in the erythrocyte and the parasite itself appears to be one of the best targets for targeted therapy since the parasite is necessarily exposed to considerable oxidative stress during its life cycle during hemoglobin degradation.

Thioredoxin reductase (TrxR) and thioredoxin (Trx) as its substrate represent the basic building blocks of the system named after them. Like mammals, *P. falciparum* has a TrxR of about 55 kDa per subunit. In contrast to Trx from other organisms, *Pf*Trx, together with the two other members of the thioredoxin superfamily, glutaredoxin, and plasmoredoxin, has an essential function in the intracellular redox balance and for the parasite's response behavior to environmental changes. The importance of the thioredoxin system for *Plasmodium* could be shown by knockout studies in which TrxR was switched off [8]. The lack of this enzyme has a lethal effect on *Plasmodium falciparum*.

Thioredoxins are ubiquitous, globular proteins with a distinct secondary and tertiary structure, known as the thioredoxin fold. The active site is exposed in a loop protruding from the protein and is characterized by the sequence motif WC(G)PC(K). The first cysteine residue exists as a thiolate anion at neutral pH, which increases the reactivity of the residue [9]. Furthermore, as a disulfide reductase, Trx has a reduction potential of approx. -270 to -290 mV [10]. Due to its exposed position, the first cysteine residue in Trx can nucleophilically attack the disulfide of a substrate protein with its thiolate anion. This creates a mixed disulfide, which is dissolved again by the second cysteine residue located 3 amino acids further inside. The intramolecular disulfide is reduced by NADPH catalyzed by the thioredoxin reductase (TrxR) in the following reaction (Fig 1).

*Plasmodium falciparum* contains three classical thioredoxins (*Pf*Trx1-3) that can be reduced by *Pf*TrxR [11], with thioredoxin-1, the member with the lowest molecular mass of 13 kDa. The NMR assignments of thioredoxin-1 of *Plasmodium falciparum* in the reduced and oxidized forms were already published [12, 13]. Preliminary solution structures were already made available by us (2MNM, 2MMO). A preliminary crystal structure of oxidized thioredoxin-1 has been deposited in the PDB database (1SYR) as well as the structure of an active site mutant of thioredoxin with thioredoxin reductase (4J56) [14].

In recent years high-pressure NMR spectroscopy has developed as a method to identify different, coexisting conformational states in proteins and to describe their dynamics (see e.g. [15, 16]). In this paper, we will discuss the differences between the two redox states in the final structures and elucidate possible structural transition by high-pressure NMR spectroscopy.

## Materials and methods

### Sample preparation

The 13 kDa protein thioredoxin-1 from *Plasmodium falciparum* (*Pf*Trx-1) was expressed and purified by us in recombinant form as previously described [12, 13] and contained a hexahistidine tag MRGIHHHHHHG at the N-terminus and thus 115 amino acids (Fig 2). For the NMR experiments, the unlabeled, the $^{15}$N, and the $^{13}$C/$^{15}$N-labeled samples were concentrated

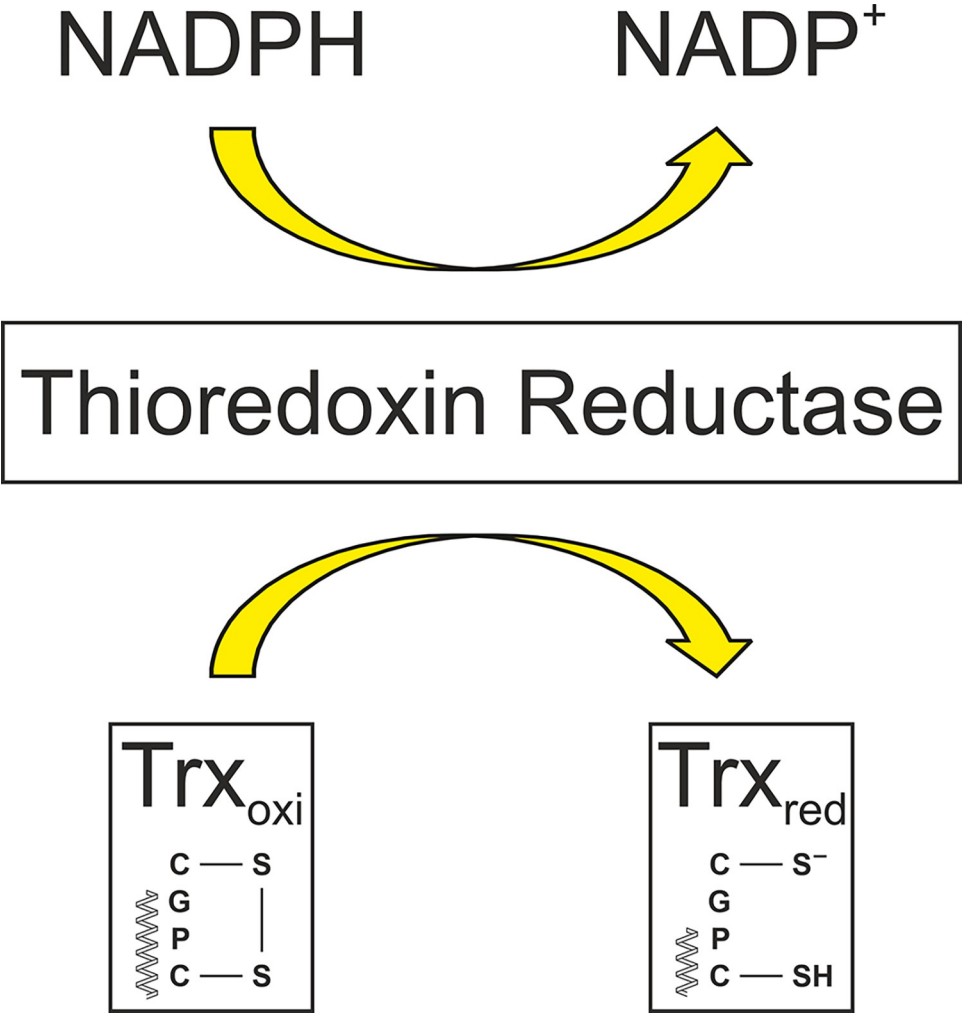

**Fig 1. General reaction scheme of thioredoxin.** Adapted from [11].

to approximately 1.0 mM using Centriprep$^{TM}$ (10 kDa MWCO) (Millipore, Darmstadt) and separated each in two batches. One batch was dialyzed against 10 mM potassium phosphate buffer, pH 7.0, in either $^1H_2O$ or $^2H_2O$ (99.5%), supplemented with 1 mM NaN$_3$, to provide the protein in the oxidized form. The same procedure was carried out for the second batch, with 1 mM DTE being added to the dialysis buffer to provide the samples in the reduced form. DSS was added to a final concentration of 0.1 mM and 5% $^2H_2O$ to the samples in $^1H_2O$. To preserve the reduced form of the reduced samples for as long as possible, they were bubbled, inside the NMR tube, with argon gas.

For the high-pressure experiments 1 mM [15]N enriched reduced and oxidized *Pf*Trx-1, respectively, was prepared analogously to the samples for structural determination. However, it was contained in 10 mM potassium phosphate buffer, pH 7.0, 0.1 mM NaN$_3$, 0.1 mM DSS, 88% H$_2$O, and 12% D$_2$O. In addition, the reduced sample contained 1 mM DTE.

### NMR spectroscopy

The NMR data for both, the reduced and the oxidized protein, were recorded on Bruker DRX-600 and DRX-800 spectrometers, equipped with cryoprobes, at a temperature of 293 K. As

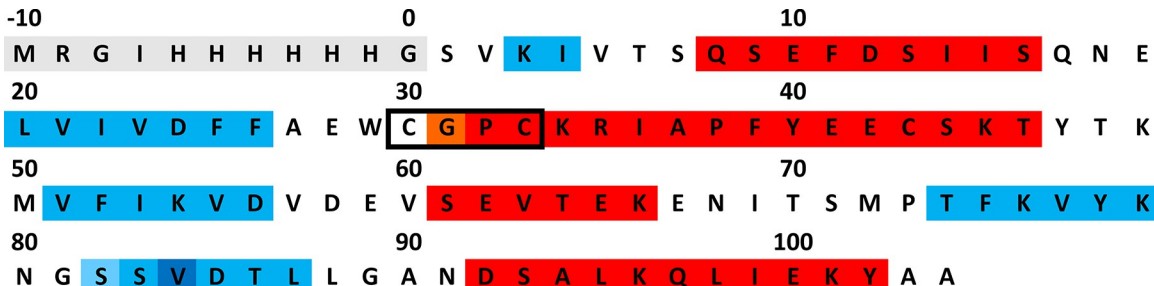

**Fig 2. Amino acid sequence of *Pf*Trx-1 used in this study.** (Dark grey) His-Tag added for purification, (red) α-helices, (light-red) residue Gly31 that extends the α-helix only in the oxidized form, (blue) β-strands, (light-blue) residue Ser82 that extends the β-strand only in the reduced form, (dark-blue) highly conserved β-bulge, (boxed) CXXC-motif. Note that in our construct the N-terminal methionine and the most commonly occurring residue Ala9 are replaced by serines.

previously described [12, 13] sequential backbone and side chain assignments were performed based on heteronuclear experiments (HNCA, CBCA(CO)NH, CBCANH, HNCO and [$^1$H-$^{15}$N]-HSQC using the samples in $^1$H$_2$O; HCCH-TOCSY and [$^1$H-$^{13}$C]-HSQC using the samples in $^2$H$_2$O). For the structure calculation, homonuclear $^1$H-NOESY and heteronuclear [$^1$H-$^{15}$N]-NOESY-HSQC spectra were recorded for both the samples in $^1$H$_2$O and $^2$H$_2$O. For the high-pressure experiments, $^1$H and [$^1$H-$^{15}$N]-HSQC spectra were obtained at 800 MHz for the reduced and oxidized $^{15}$N-enriched samples. The proton shifts were referenced using the $^1$H resonance frequency of the methyl group of DSS, and the $^{13}$C and $^{15}$N resonances were indirectly calibrated by the IUPAC recommendations [17]. The data were processed using the program Topspin (Bruker, Karlsruhe) and evaluated using the program AUREMOL [18].

High-pressure NMR experiments were performed in borosilicate capillaries (inner diameter of 1 mm, outer diameter of 4 mm) in the pressure range between 3 MPa and 200 MPa in steps of 20 MPa as described by [19, 20]. Measurements were performed at 800 MHz at 293 K. Typical measurement time per pressure point was 47 min.

## Sequential assignments

We have already published the complete assignment of the protein in the reduced and oxidized form [12, 13].

## Structure calculation

Experimental distance restraints were acquired by analyzing the 2D- and 3D-NOESY spectra. Correlation peaks were picked, assigned and integrated using the program AUREMOL [18]. The normalization of the distances obtained from the volumes was performed using a series of unambiguous peaks and referring to fixed interatomic distances. Experimental restraints of dihedral angles were obtained from the chemical shifts $^{15}$N$^H$, $^{13}$C$^\alpha$, $^{13}$C$^\beta$, and $^{13}$C' using the program TALOS+ [21].

The structural calculation was performed using a simulated annealing protocol and the CNS 1.1 program [22]. Distance and dihedral angle restraints were included, as well as the $^1$H chemical shifts of the loop around the active site (residues 27 to 35) as implemented in CNS. The residues of the His-Tag were not included in the structure calculations. A total number of 1000 structures were calculated for each reduced and oxidized *Pf*Trx-1.

The ω-angles in front of proline residues were determined from the $^{13}$C-chemical shift differences $\Delta_{\beta\gamma}$ of the C$^\beta$ and C$^\gamma$ shifts of the proline residues according to Schubert et al. [23] and/or from the NOEs between the amino acid X preceding the proline residue and the proline

residue according to Wüthrich et al. [24]. A $\Delta\beta\gamma$ from 0.0 ppm to 4.8 ppm the peptide bond conformation is predicted to be 100% *trans*, whereas from 9.15 ppm to 14.4 ppm to be 100% *cis*. All peptide bonds were restricted to be planar and *trans* except when clear experimental evidence for a *cis*-bond was available.

## Evaluation of high-pressure NMR data

The chemical shift values of the amide $^1$H and $^{15}$N atoms obtained from the [$^1$H-$^{15}$N]-HSQC spectra as a function of pressure were first corrected for random coil pressure effects by subtracting the known pressure dependence of the amino acid X in the model peptide Ac-Gly-Gly-X-Ala-NH$_2$ [25]. A second correction, taking into account the neighbor's contribution [26] was performed. Combined chemical shifts $\delta_{\text{comb}}$ of each residue were calculated from $\delta_{1\text{H}}$ and $\delta_{15\text{N}}$ using amino acid and atom-specific factors [27]. Because of the required additivity of the chemical shifts the Euclidean distance has been used in these calculations.

As previously described by Kremer et al. [28] the obtained chemical shifts $\delta$ ($\delta_{1\text{H}}$, $\delta_{15\text{N}}$, and $\delta_{\text{comb}}$) at the temperature $T_0$ were then fitted as a function of the pressure $P$:

$$\delta(P, T_0) = \delta_0(P_0, T_0) + B_1(P - P_0) + B_2(P - P_0)^2, \qquad [\text{Eq (1)}]$$

where $\delta_0$ is the chemical shift at ambient pressure $P_0$ and $B_1$ and $B_2$ are the first and second-order pressure coefficients.

If the protein has $n$ states $i$, ($i = 1, n$) in equilibrium at temperature $T$, coexisting under conditions of fast exchange $|\Delta\omega_{ij}\tau_c| << 1$, the chemical shift $\delta$ observed experimentally will result from a combination of the chemical shifts $\delta_i$ of each of these states:

$$\delta = \frac{\delta_1 + \sum_{i=2}^n \delta_i exp(-\frac{\Delta G_{1i}}{\text{R}T})}{1 + \sum_{i=2}^n exp(-\frac{\Delta G_{1i}}{\text{R}T})}, \qquad [\text{Eq (2)}]$$

where 1 is an arbitrarily selected state, $\Delta G_{1i}$ the Gibbs free energy, and R the molar gas constant. The Gibbs free energy is given as [29]:

$$\Delta G_{1i}(T_0, P) = \Delta G_{1i}^0(T_0, P_0) + \Delta V_{1i}^0(P - P_0) - \frac{\Delta\beta_{1i}^{'0}}{2}(P - P_0)^2 = \text{R}Tln(K_{1i}), \qquad [\text{Eq (3)}]$$

where $\Delta V^0_{1i}$, $\Delta\beta^{'0}_{1i}$ are the differences of the partial molar volumes $V^0$ and the partial molar compressibility factors $\beta^{'0}$ (-$\partial$V/$\partial$P) between states 1 and $i$ at temperature $T_0$ and pressure $P_0$, respectively, and $K_{1i} = [1]/[i]$ the equilibrium constant. Finally, the relative population $p_i$ can be calculated as

$$p_i(P, T_0) = \frac{K_{1i}(P, T_0)}{\sum_{j=1}^N K_{1j}(P, T_0)}. \qquad [\text{Eq (4)}]$$

The experimental data were fitted to Eq 2 in an iterative way similar to the method described earlier by Munte et al. [20] for high-pressure spectroscopy on A$\beta$ (1–40). A three-state model has been assumed.

## Representation of structures, estimation of surface accessibility, and interacting residues

The representation of 3D structures and protein surfaces as well as the estimation of the water-accessible surface (using a rolling sphere with a diameter of 0.14 nm) were performed with MolMol [30]. Interacting residues were identified as described by Schumann et al. [27]. The

**Table 1. Completeness of resonance assignments in the reduced and oxidized *Pf*Trx-1 at pH 7.0 and 293 K.** From [12, 13].

| | Reduced | Oxidized |
|---|---|---|
| Assigned chemical shifts | | |
| $^1H^N$, $^{13}C^\alpha$ and $^{15}N^H$ backbone [a] | 99% | 99% |
| $^{13}C'$ | 89% | 96% |
| $^1H^\alpha$ | 99% | 99% |
| all side-chain atoms [b] | 88% | 90% |
| BioMagResBank accession number | 6442 | 16147 |

[a] excluding the proline residues.

[b] except non-protonated $^{13}C$ shifts of the aromatic rings.

structures of the reduced and the oxidized *Pf*Trx-1 are deposited in the Protein Data Bank under accession codes 2MMN and 2MMO, respectively.

The obtained 10 lowest energy NMR-structures of oxidized as well as reduced *Pf*Trx were deposited in the PDB database with the PDB-IDs 2MMN and 2MMO.

## Results

### Solution structure of reduced and oxidized *Pf*Trx-1

Nearly complete backbone and side chain assignments were obtained for both reduced and oxidized *Pf*Trx-1 (Table 1). Almost all resonances assigned in the reduced form could also be assigned in the oxidized form in the matched NMR samples.

As mentioned in Materials and Methods all peptide bonds were assumed to occur in the *trans*-configuration. An exception is the peptide bonds in front of proline residues that are often found in *cis*-configuration. *Pf*Trx-1 contains three proline residues, Pro31, Pro38, and Pro73. The strong $H^\alpha$–$H^\alpha$ NOE between Met72 and Pro73 unambiguously defined the *cis*-peptide bond of the Pro73 in oxidized and reduced *Pf*Trx-1 (Table 2). For the two other proline residues, the strong $H^N$-$H^\delta$ sequential NOE and a $\Delta\beta\gamma < 4.8$ ppm are indicative of a *trans*-peptide bond.

**Table 2. Conformation of the peptide bond in front of proline residues[a].**

| Amino Acid | $C^\beta$ (ppm) | $C^\gamma$ (ppm) | $\Delta\beta\gamma$ (ppm) | Strong NOE | Isomer |
|---|---|---|---|---|---|
| Reduced | | | | | |
| P32 | 32.38 | 28.05 | 4.93 | - [b] | *Trans* |
| P38 | 30.46 | 29.09 | 1.37 | $H^N(37)$-$H^\delta(38)$ | *Trans* |
| P73 | 33.47 | - [c] | - | $H^\alpha(72)$-$H^\alpha(73)$ | *Cis* |
| Oxidized | | | | | |
| P32 | 32.43 | 28.05 | 4.38 | - [b] | *Trans* |
| P38 | 30.52 | 28.05 | 2.47 | $H^N(37)$-$H^\delta(38)$ | *Trans* |
| P73 | 34.07 | - [c] | - | $H^\alpha(72)$-$H^\alpha(73)$ | *Cis* |

[a] The peptide bond conformation was determined from the chemical shift difference $\Delta\beta\gamma$ of the $C^\beta$ and $C^\gamma$ shifts of the proline residues according to [23] and/or from the NOEs between the amino acid X preceding the proline residue and the proline residue according to [24].

[b] No unambiguous NOEs could be found.

[c] Resonance could not be assigned.

**Table 3. NMR restraints for molecular dynamics and simulated annealing of the reduced and oxidized *Pf*Trx-1 at pH 7.0 and 293 K[a].**

| Restraints | Reduced | Oxidized |
|---|---|---|
| Distances (NOEs) | 2967 | 2966 |
| Intraresidual ($i$, $i$) | 725 | 725 |
| Sequential ($i$, $i$+1) | 698 | 698 |
| Intermediate-range ($i$, $i$+$j$; $1 < j \leq 4$) | 564 | 563 |
| Long range ($i$, $i$+$j$; $4 < j$) | 980 | 980 |
| Dihedral angles | 180 | 180 |
| Proton chemical shifts [b] | 180 | 180 |
| Number of restraints/residue | 32 | 32 |

[a] The His-Tag has been omitted from the calculations and the statistics.

[b] The chemical shifts of all protons of the active site loop (residues 27 to 35) were used as restraints as implemented in CNS.

Table 3 summarizes the restraints used for the structure calculations in the two forms. In total 3327 (3326 for the oxidized protein) structural restraints were available for the calculations.

Among the 1000 structures obtained, the 10 with the lowest total energy were selected for analysis. Table 4 shows statistics of the structures obtained.

The structures of the reduced and the oxidized proteins are depicted in Fig 3. The calculated, lowest energy structures show an excellent structural similarity, reflecting both the excellent quality of the spectra and the number of NOEs assigned. Consistent with chemical shift analysis and the NOE patterns, a $\beta_1\ \alpha_1\ \beta_2\ \alpha_2\ \beta_3\ \alpha_3\ \beta_4\ \beta_5\ \alpha_4$ topology is obtained. This topology is typical for thioredoxins: a central β-sheet formed by 5 β-strands, surrounded by 4 α-helices. The helices encompass residues 8–16 ($\alpha_1$), 32–46 ($\alpha_2$) for the reduced form or 31–46 ($\alpha_2$) for the oxidized form, 61–66 ($\alpha_3$) and 92–102 ($\alpha_4$). The β-sheet has the pattern ↑$\beta_1$ ↑$\beta_3$ ↑$\beta_2$ ↓$\beta_4$ ↑$\beta_5$ and comprises residues 3–4 ($\beta_1$), 20–26 ($\beta_2$), 51–56 ($\beta_3$), 74–79 ($\beta_4$) and 82–88 ($\beta_5$) for the

**Table 4. Structural statistics of the 10 lowest energy structures (from 1000) of the reduced and oxidized *Pf*Trx-1 at pH 7.0 and 293 K.**

| | Reduced | Oxidized |
|---|---|---|
| Energy (kJ mol$^{-1}$) [a] | | |
| Total | 2211 (38) | 8766 (32) |
| Bonds | 107.5 (4.4) | 950.7 (5.2) |
| Angles | 622 (12) | 1675 (23) |
| Impropers | 174.7 (7.0) | 445.9 (7.1) |
| Van der Waals | 424 (24) | 453 (22) |
| NOEs | 422 (10) | 4300 (38) |
| Protons [b] | 307 (24) | 533 (16) |
| Dihedral | 153 (14) | 410 (10) |
| RMSDs (Å) | | |
| Main chain atoms [c] | 0.300 | 0.266 |
| All atoms | 1.008 | 0.981 |

[a] Value in parentheses: standard deviation.

[b] Chemical shift energy of all protons of the active site loop (residues 27 to 35) as implemented in CNS.

[c] All backbone N$^H$, C$^\alpha$, and C' atoms.

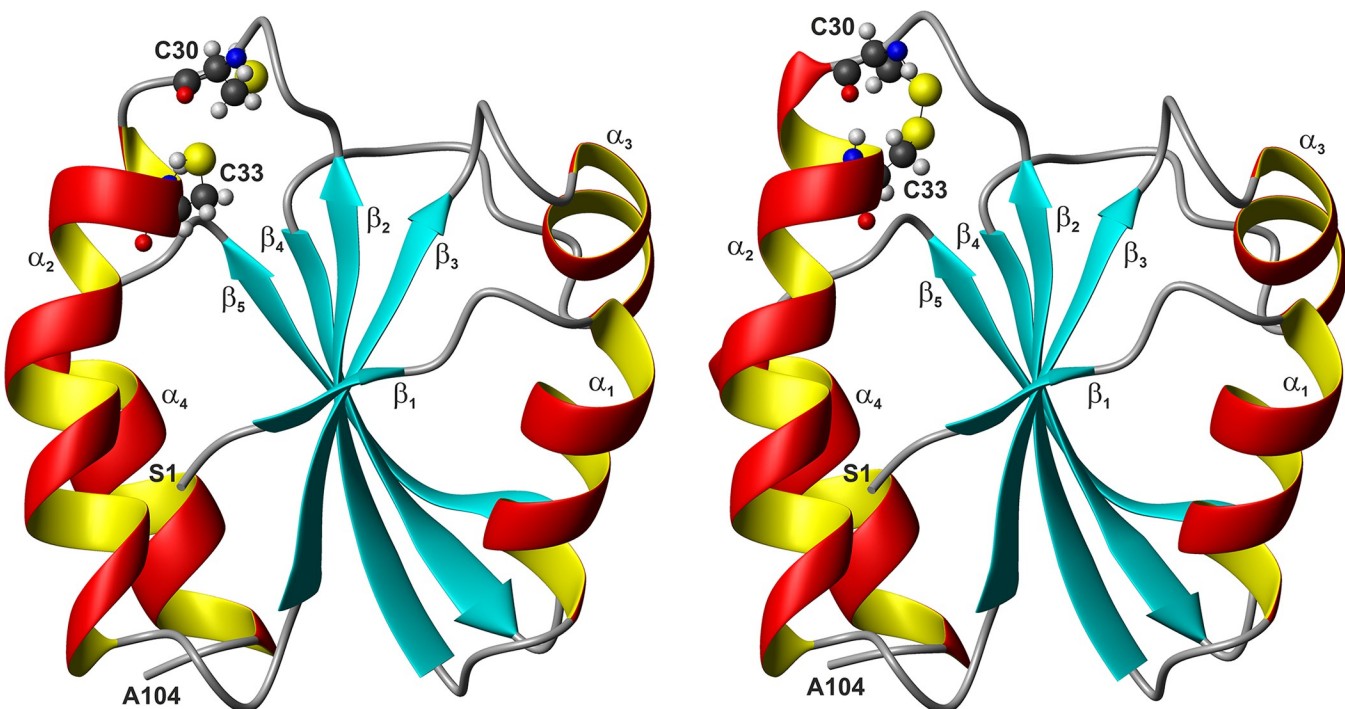

**Fig 3. NMR structure of reduced and oxidized thioredoxin from *Plasmodium falciparum*.** Lowest energy structures in the ribbon representation, with atomic details of the CGPC active site and showing (**left**) the two cysteines in their reduced form or (**right**) the two cystines in the oxidized form. The disordered N-terminal His-Tag is not presented. The sulfur atoms are shown in yellow.

reduced form or 83–88 ($\beta_5$) for the oxidized form. This is in good accordance with the TALOS + [21] predictions. For the $\alpha$-helices $\alpha_1$ and $\alpha_4$ and the $\beta$-strand $\beta_4$, the same residues as in the 3D structure are predicted. TALOS+ predicts for the remaining secondary structure elements slightly larger ranges: $\alpha_2$, residues 32–47 in the reduced form or 32–45 in the oxidized form, $\alpha_3$, residues 58–66, $\beta_1$, residues 2–5, $\beta_2$, residues 20–27 in the reduced form or 20–26 in the oxidized form, $\beta_3$, residues 50–55, $\beta_4$, residues 74–79, $\beta_5$ residues 85–88 in the reduced form or 84–88 in the oxidized form. As with other thioredoxins, the Val-84 residue forms a classic anti-parallel bulge in the $\beta_5$ stand (type AC) [31]: the conformation of residue 84 is nearly $\alpha_R$-helical, residues 84 and 85 are close to the extended $\beta$ conformation, residues 77, 84 and 85 have their side chains pointing to the same direction of the $\beta$-sheet. The active site motif Cys-Gly-Pro-Cys is located at the beginning of the $\alpha_2$ helix and the cysteines are already positioned so that only a slight change in their $\chi$ angles allows the switch between reduced and oxidized form.

The 10 lowest energy structures of the reduced *Pf*Trx-1 were analyzed using the program PROCHECK-NMR [32]. The Ramachandran Plot for these structures resulted in 85.8% of the backbone angles in the most favored regions, 13.2% in the additional allowed regions, 0.9% in generously allowed regions, and 0.0% in disallowed regions. The equivalent resolutions were 1.8 Å based on the Ramachandran plot quality assessment, 2.4 Å on the hydrogen bond energies, 3.0 Å on the $\chi_1$-pooled standard deviation, and 4.0 Å on the dihedral angles G-factor (see also S1 and S2 Files). The solution structures of the *Plasmodium falciparum* thioredoxin-1, in the oxidized and reduced form, are stored in the PDB database (2MMN, 2MMO).

Fig 4 shows the sequence dependence of the root-mean-square differences between the structures of the reduced and oxidized *Pf*Trx-1. The main structural differences above $\sigma_0$ (28

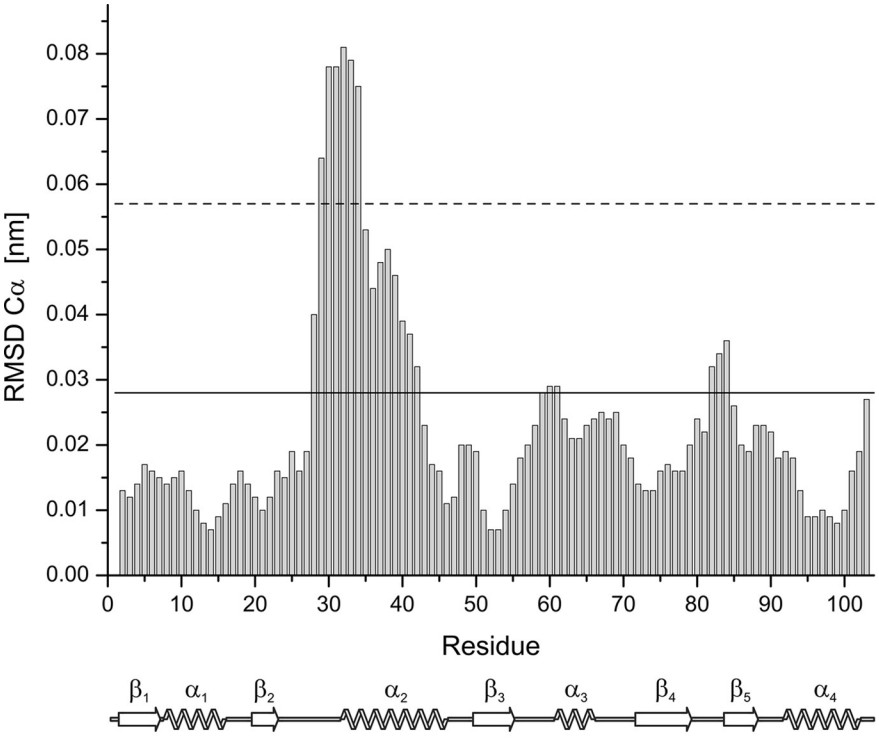

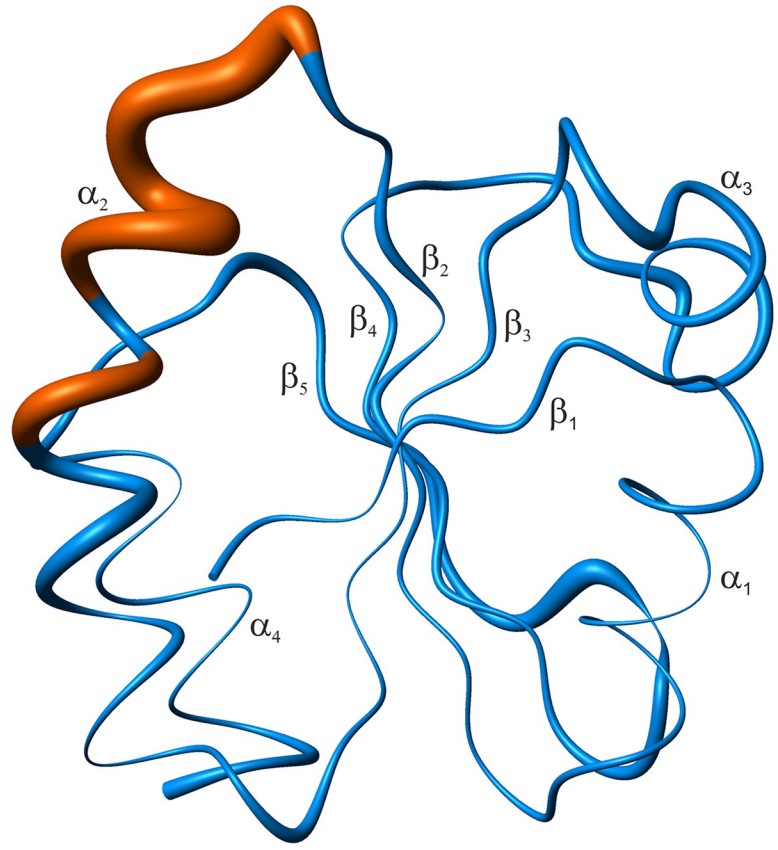

**Fig 4. RMSD-differences between the structures of the reduced and the oxidized thioredoxin from *Plasmodium falciparum*.** (top) The local RMSD is calculated for the $C^{\alpha}$ atoms of the lowest energy structures. Values above the reduced standard deviation to zero $\sigma_0^{red}$ of 0.028 nm (solid line) and $2\sigma_0^{red}$ of 0.057 nm (dashed line) are considered to show significant conformation differences [27]. (bottom) "Sausage" representation of the differences in the structures of the reduced and the oxidized *Pf*Trx-1. The tubes represent the plot of the local RMSD values for the $C^{\alpha}$ atoms of the 2 structures, reduced and oxidized *Pf*Trx-1. The diameter of the tubes is proportional to the RMSD, with values $\leq 0.05$ nm in blue and values $> 0.05$ nm in dark orange.

to 43, 48 to 49; 57 to 70; 79 to 86, 88 to 90) are located around the CXXC-motif including the adjacent helix $\alpha_2$, helix $\alpha_3$ and the adjacent loop, and β-strand $\beta_5$.

Fig 5 depicts the changes in the surface of reduced thioredoxin after oxidation and shows the residues that change their position significantly. They may be used by interacting proteins such as thioredoxin reductase (TrxR) or glutathione reductase (GR) for recognizing the oxidation state of thioredoxin. Thioredoxin reductase is a homo-dimer in the crystal structure [14] that interacts differently with two Trx-molecules called in the following Trx@site1 and Trx@site2. Their residues interacting with TrxR were identified as described by Schumann et al. [27]. Only a part of the residues that form different surface structures in the reduced and oxidized form of thioredoxin are involved in the interaction with thioredoxin reductase (Fig 5). The stretch of residues at the back of thioredoxin (K79 to A90) that have different positions in the two forms are not directly involved in the interaction with thioredoxin reductase (see Discussion).

The solution structures in the oxidized and reduced form have been stored in the PDB database with the PDB-IDs 2MMN and 2MMO.

## Pressure response of thioredoxin in reduced and oxidized form

We recorded [$^1$H-$^{15}$N]-HSQC spectra of reduced and oxidized thioredoxin at 293 K in the pressure range from 0.1 MPa to 200 MPa. The pressure dependence of $^1$H and $^{15}$N chemical shifts of the cross-peaks were described by a polynomial of the second degree (Eq 1). The chemical shifts were corrected for random-coil effects using the pressure response of model peptides [25] and for the contribution of the neighbored amino acids [26].

Fig 6 represents the combined random-coil corrected first and second-order pressure coefficients $B_1^*$ and $B_2^*$. In addition, the ratio of $B_2/B_1$ of the combined uncorrected values is presented that is related to the local compressibility [33]. The mean values and standard deviations relative to their means are indicated in the plots; residues that behave like residues in random-coil model peptides would have pressure coefficients $B_1^* = B_2^* = 0$.

It is apparent that most residues have combined $B_1^*$ and $B_2^*$ values different from 0 and thus show a pressure response different from the random coil structures (Fig 6). Mostly, they are similar in reduced and oxidized *Pf*Trx-1 but some amino acids show clear differences. The mean $B_1^*$ values of reduced and oxidized thioredoxin are significantly different with -0.145 and -0.114 ppm $GPa^{-1}$, respectively. The mean $B_2^*$ values in the reduced and oxidized form are 0.179 and 0.119 ppm $GPa^{-2}$, respectively. The mean ratios of the pressure coefficients $B_2/B_1$ are -0.484 and -0.831 $GPa^{-1}$ in the reduced and oxidized form respectively. They strongly differ at some points in the structure after the formation of the disulfide bond between C30 and C33 (Fig 6). The residues with very high or very low $B_2/B_1$ ratios are also depicted on the surface of reduced and oxidized *Pf*Trx-1 (Fig 7). Clear differences between the two oxidation states of the protein become visible.

A thermodynamical analysis can be performed when the pressure dependence of chemical shifts is fitted with a suitable method (see Materials and Methods). Typical pressure responses are represented by monophasic or biphasic curves. Fig 8 shows such an example for E28 where

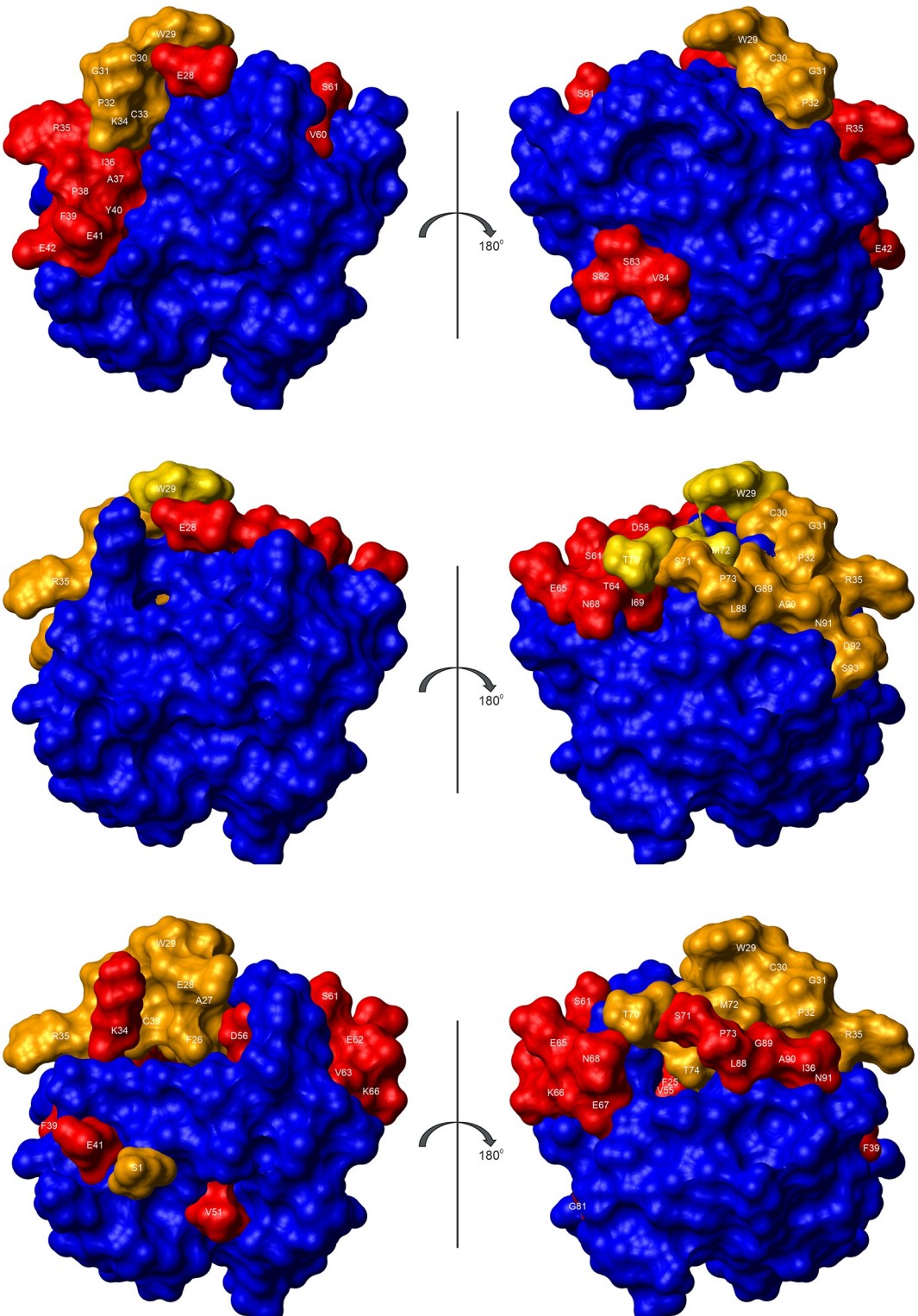

**Fig 5. Changes in the surface of thioredoxin after oxidation of the CXXC motif.** (top) RMSD between the 3D-structures of reduced and oxidized thioredoxin: deviations $\geq 2\sigma_0^{red}$ (orange), $\geq \sigma_0^{red}$ and $< 2\sigma_0^{red}$ (red), $< \sigma_0^{red}$ (blue). For details see also Fig 3. (middle) TrxR is a dimer that binds two Trx (Trx@site1 and Trx@site2) at two different sites: interaction of Trx@site1 with TrxR (orange), of Trx@site2 with TrxR (red), of Trx@site1 and Trx@site2 with TrxR (yellow). (bottom) The oxidation of Trx induces chemical shifts changes $\Delta\delta_{comb}$, calculated over all assigned atoms $^1$H, $^{13}$C, and $^{15}$N and combined [27]: changes $\geq 2\sigma_0^{red}$ (orange), $\geq \sigma_0^{red}$ and $< 2\sigma_0^{red}$ (red), $< \sigma_0^{red}$ (blue).

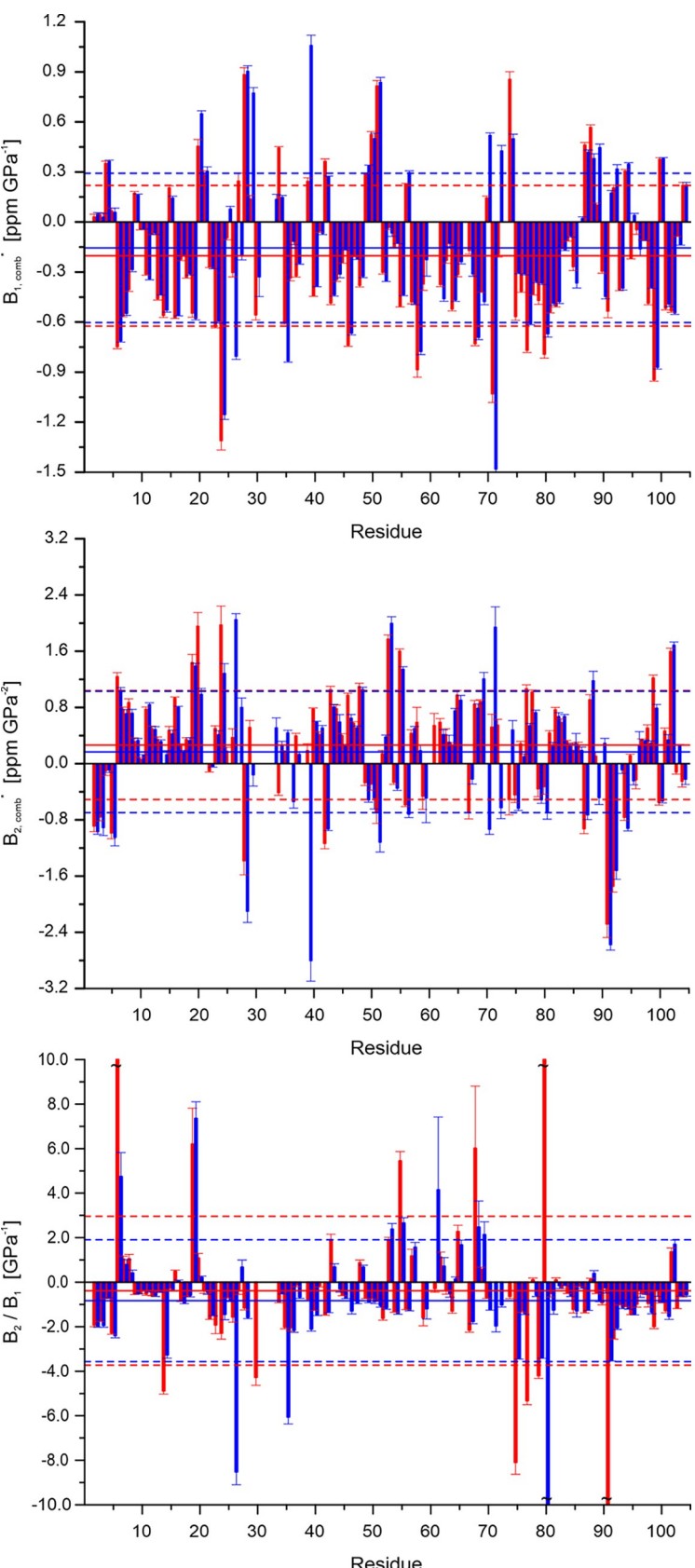

**Fig 6. Pressure coefficients of $^{15}$N enriched thioredoxin in the reduced and oxidized state.** The combined H$^N$ and N$^H$ pressure coefficients $B_1$*and $B_2$*, corrected for the random-coil effect, and the $B_2/B_1$ values are plotted as a function of the residue number. (Red) reduced *Pf*Trx-1, (blue) oxidized *Pf*Trx-1. Solid lines, mean values; broken lines, standard deviations σ. Note that values where the experimental error was larger than the value itself were omitted in the calculation of the mean. Temperature 293 K. 1 mM $^{15}$N *Pf*Trx-1 in 10 mM potassium phosphate buffer, pH 7.0, 0.1 mM NaN$_3$, 0.1 mM DSS, 88% H$_2$O, 12% D$_2$O. In addition, the reduced sample contained 1 mM DTE. For details of sample preparation see Materials and Methods.

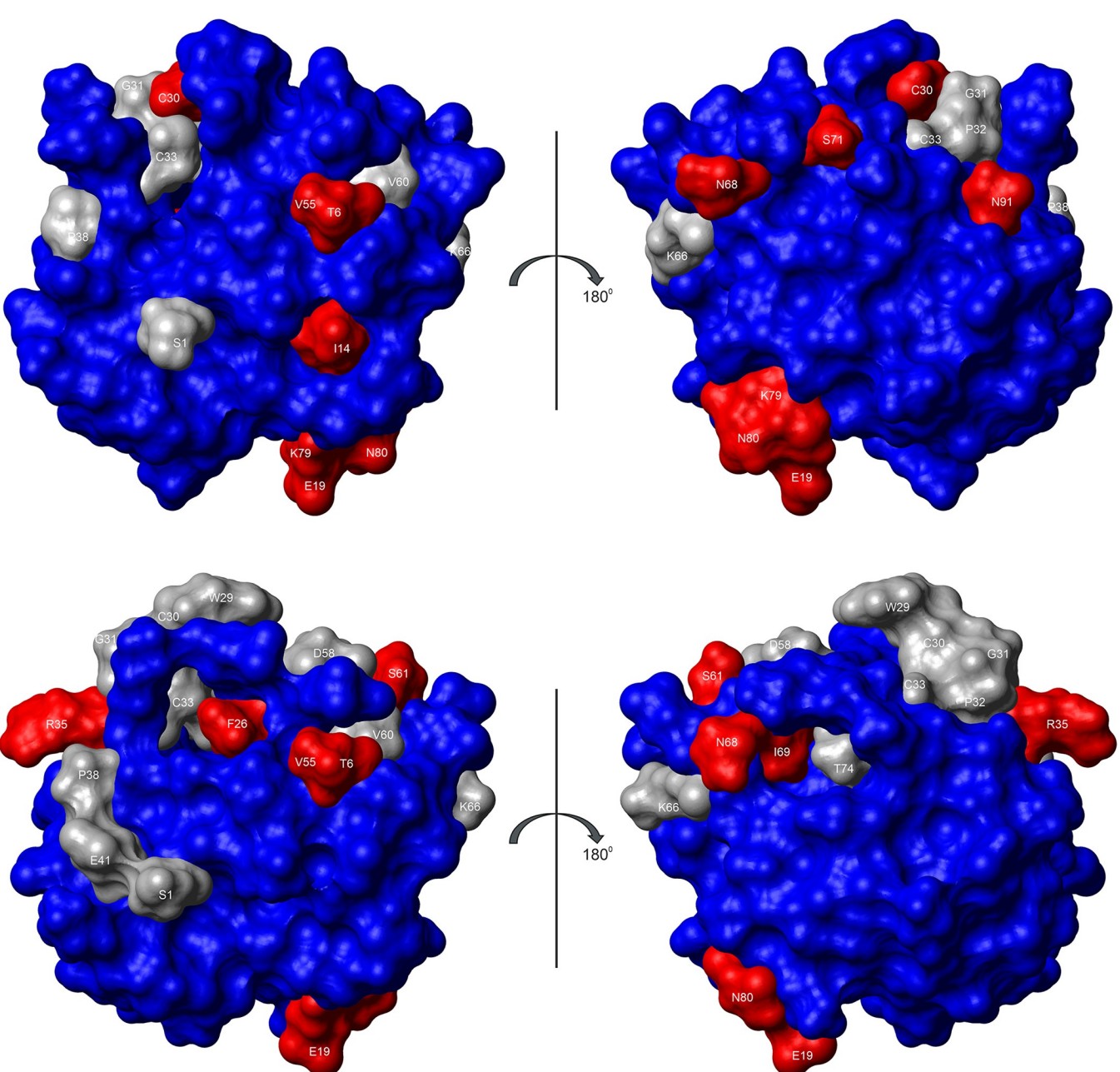

**Fig 7. Pressure response in reduced and oxidized $^{15}$N enriched thioredoxin.** The combined $^1$H and $^{15}$N $B_2/B_1$ values are represented on the surface of reduced (top) and oxidized (bottom) thioredoxin. (Blue) residues with $(<B_2/B_1>—σ) < B_2/B_1 < (<B_2/B_1> + σ)$, (red) with $B_2/B_1 ≥ (<B_2/B_1> + σ)$ or $B_2/B_1 ≤ (<B_2/B_1>—σ)$, (grey) no values available. Temperature 293 K. Sample composition see Fig 6.

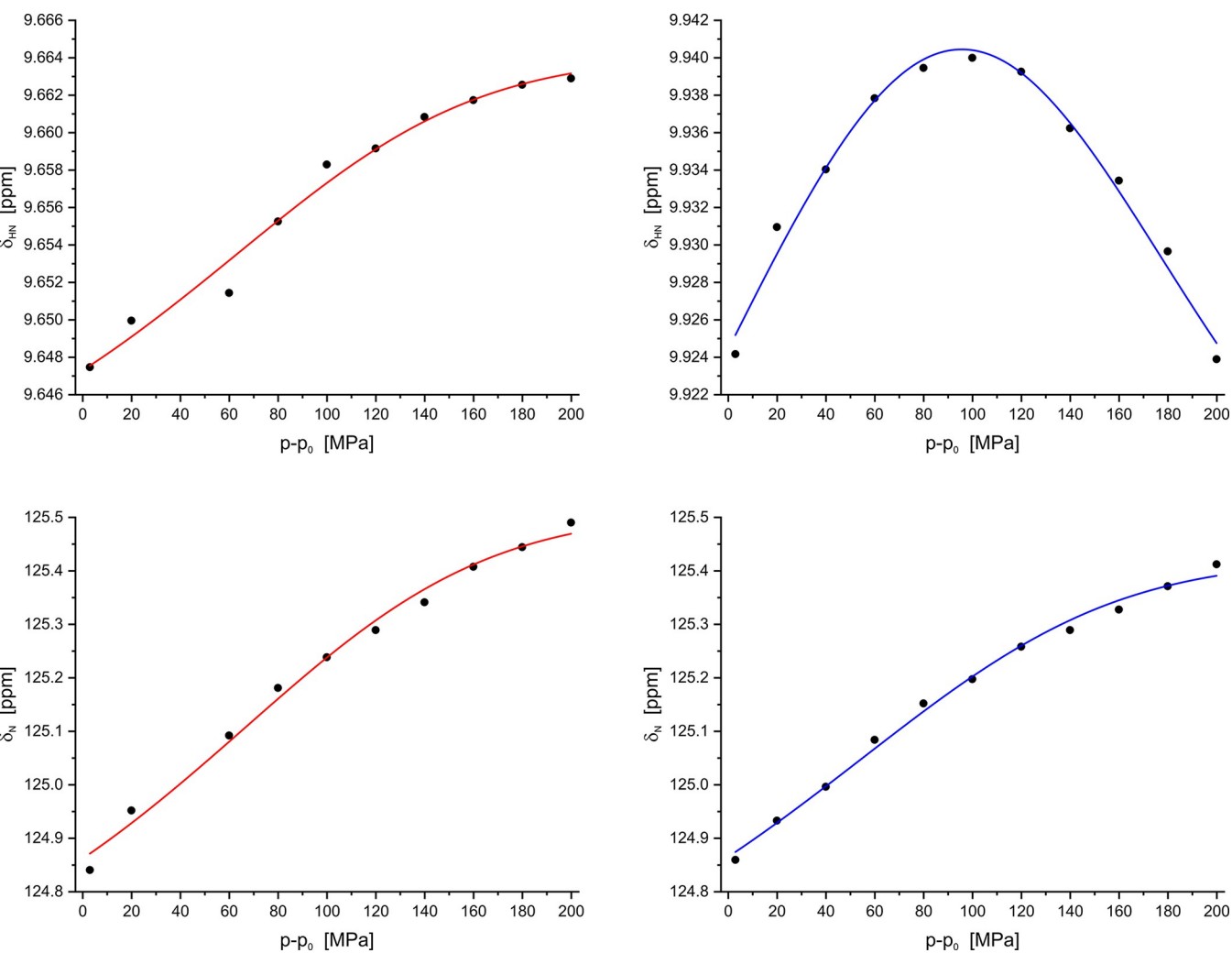

**Fig 8. Characteristic pressure-dependent chemical shift changes in $^{15}$N enriched thioredoxin.** The chemical shifts of the amide H$^N$ and N$^H$ atoms of residue E28 are depicted together with the fit curves calculated (Eqs 2, 3) with the thermodynamical parameters presented in Table 5. (blue) oxidized *Pf*Trx-1, (red) reduced *Pf*Trx-1. Temperature 293 K. Sample composition see Fig 6.

**Table 5. Thermodynamical parameters of reduced and oxidized $^{15}$N enriched *Pf*Trx-1 at pH 7.0 and 293 K[a].**

| State $i$ | $p_i$ [b] | Transition | $\Delta G^0$ [kJ mol$^{-1}$] | $\Delta V^0$ [mL mol$^{-1}$] | $\Delta \beta'^0$ [mL MPa$^{-1}$ mol$^{-1}$] | $\Delta \beta'^0 / \Delta V^0$ [GPa$^{-1}$] |
|---|---|---|---|---|---|---|
| Reduced | | | | | | |
| $1_{re}$ | 0.711 | 1–2 | 2.81 ± 0.21 | -28.69 ± 0.99 | -0.0488 ± 0.0085 | 1.70 |
| $2_{re}$ | 0.225 | 2–3 | 3.06 ± 0.56 | -34.3 ± 2.9 | 0.026 ± 0.035 | -0.75 |
| $3_{re}$ | 0.064 | 1–3 | 5.87 ± 0.52 | -63.0 ± 2.7 | -0.022 ± 0.034 | 0.35 |
| Oxidized | | | | | | |
| $1_{ox}$ | 0.684 | 1–2 | 2.46 ± 0.19 | -28.41 ± 0.99 | -0.0540 ± 0.0088 | 1.90 |
| $2_{ox}$ | 0.249 | 2–3 | 3.21 ± 0.78 | -32.15 ± 3.5 | 0.038 ± 0.029 | -1.18 |
| $3_{ox}$ | 0.067 | 1–3 | 5.67 ± 0.59 | -60.6 ± 2.5 | -0.016 ± 0.028 | 0.26 |

[a] Temperature 293 K. Sample composition see Fig 6. The errors given represent standard errors.

[b] $p_i$, probability of state $i$ at 293 K and 0.1 MPa.

the pressure dependence of the amide proton in the oxidized form clearly shows a biphasic behavior that is not observed in the reduced form of the protein. The formal description of biphasic curves requires the assumption of at least three conformational states whose populations change with pressure. All experimental curves could be perfectly fitted assuming a three-state model with the parameters given in Table 5. This represents a consistent but not unique parameter set explaining the experimental data set (see Discussion). From the free energies, the probabilities $p_i$ to find state $i$ can be calculated for any pressure at 293 K (Eq 4). The values for ambient pressure are given in Table 5.

Larger chemical shift changes are usually (but not necessarily) correlated with larger structural changes. The individual chemical shift changes associated with the two transitions 1–2 and 2–3 are directly obtained from the fit of the pressure-dependent chemical shifts with the parameters represented in Table 5. They are represented graphically in the *Pf*Trx-1 structures in Fig 9. If the shift changes associated with the first transition are at least two times larger than those associated with the second transition, they are plotted in red. In the opposite case, they are plotted in green. In the intermediary cases, they are plotted in blue, if the corresponding amino acids were not observed in the pressure study they are depicted in grey. Qualitatively, the first transition seems to prevail in the helical regions, the second in the β-sheet regions.

## Discussion

### Solution structure of reduced and oxidized *Pf*Trx-1

*Pf*Trx-1 contains a pair of active site cysteine residues that form transient disulfide bridges during the redox reaction. Previously, we had fully assigned the spin systems of the reduced [12] and the oxidized protein [13] using multidimensional NMR spectroscopy on uniformly $^{13}$C - and $^{15}$N-enriched proteins. The canonical secondary structure elements in both structures were predicted from the heteronuclear chemical shifts and, as to be expected, are not significantly altered by the formation of the disulfide bond. They correspond closely to the secondary structures found in the three-dimensional structures determined here (Fig 3).

The reduced and the oxidized protein both show the typical expanded thioredoxin fold with 5 β-strands surrounded by 4 α-helices in a $\beta_1$ $\alpha_1$ $\beta_2$ $\alpha_2$ $\beta_3$ $\alpha_3$ $\beta_4$ $\beta_5$ $\alpha_4$ topology (Fig 3). Note, that the thioredoxin fold itself does not contain $\beta_1$ and $\alpha_1$ and consists of 4 β-sheets and 3 α-helices [34]. Like other thioredoxins, Val-84 forms a classic antiparallel bulge in the $\beta_5$ stand (type AC) [31].

The $^1$H and $^{15}$N signals of the side chain amides of Asn18 in the reduced as well as in the oxidized form of thioredoxin are strongly downfield shifted by approximately 0.95 ppm ($H^{\delta 21}$), 0.63 ppm $H^{\delta 22}$, and 5.7 ppm ($N^\delta$) from their random coil position. There is no aromatic residue in close contact in the 3D structures that could explain these large downfield shifts. The shift may arise from the dipole moment of helix $\alpha_1$ itself.

### Structural changes induced by disulfide bond formation

As to be expected the formation of the disulfide bond between C30 and C33 leads to strong local structural changes (Fig 4) without changing the overall fold. However, it induces also distinct long-range conformational changes. Fig 4 shows the sequence-dependent structural changes measured by the $C^\alpha$-RMSD above the standard deviation to zero $\sigma_0$ of 0.028 nm. They encompass the CGPC motif itself and most of the adjacent helix $\alpha_2$, β-strand $\beta_5$ and its connecting loops, and the N-terminal part of helix $\alpha_3$.

Thioredoxin interacts with several different proteins that have to recognize the oxidation state of the protein. For such a recognition the structural changes at the surface are most important. They are depicted in Fig 5 which shows that mainly surface residues around the

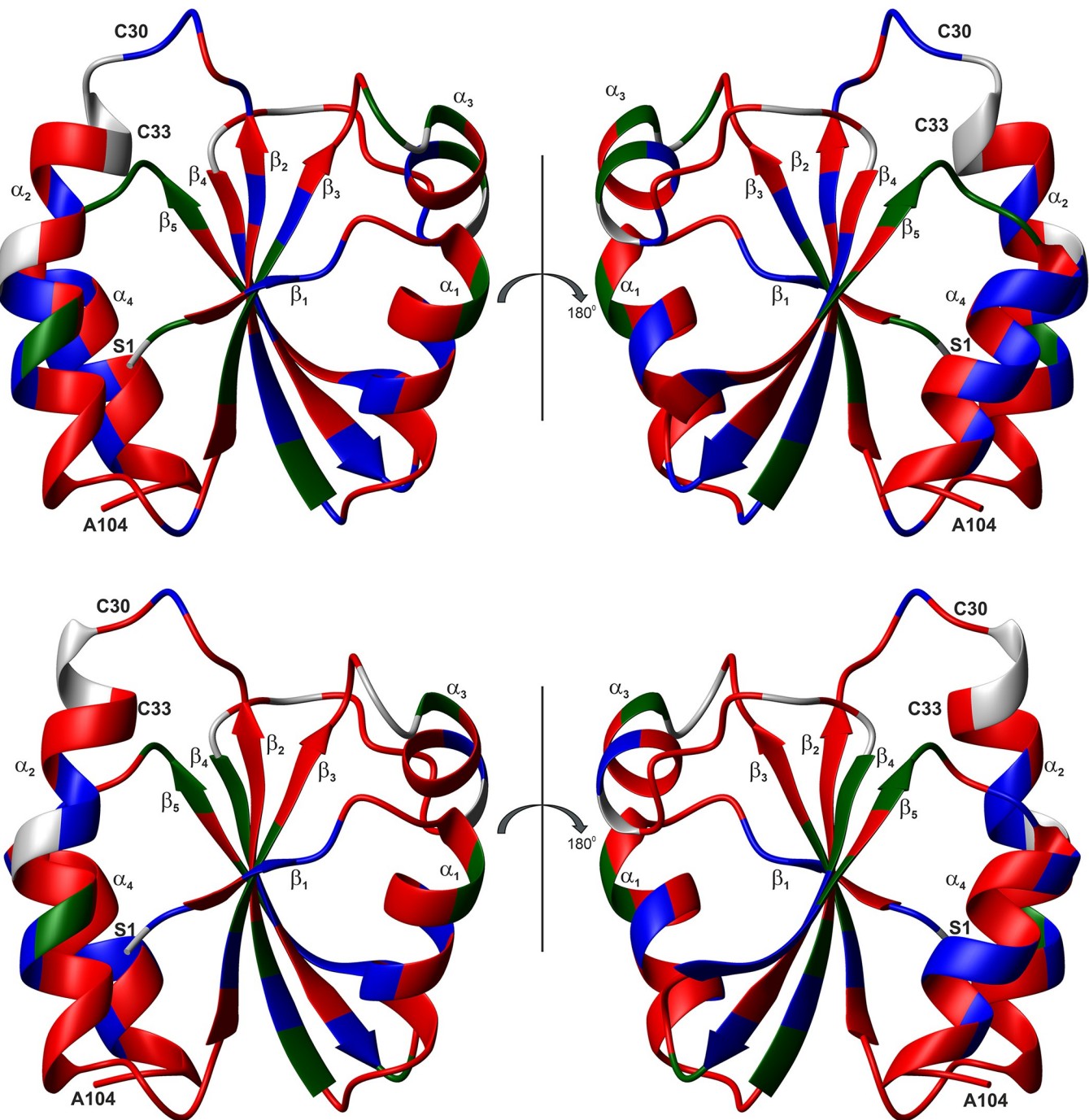

**Fig 9. Regions with dominant contributions of the two main structural transitions.** (Top) reduced *Pf*Trx-1, (bottom) oxidized *Pf*Trx-1. (Grey) no data available, (blue) no dominant transition according to the shift differences of the three states defined. At least for one nucleus (red) $|\delta_2 - \delta_1| \geq 2\,|\delta_3 - \delta_2|$ or (green) $|\delta_3 - \delta_2| > 2\,|\delta_2 - \delta_1|$.

redox center and the subsequent helix α2 (E28 to E42) are influenced but also a group of residues very distant from the redox center (S82, S83, and V84 of the β-bulge of β-strand $\beta_5$).

Chemical shift changes can be measured with high accuracy and often show structural changes with higher sensitivity than three-dimensional structures themselves. Since oxidized

and reduced *Pf*Trx-1 were almost completely assigned most of the $^{1}$H, $^{13}$C, and $^{15}$N chemical shifts are available. We calculated the amino acid and atom-specific combination of all known chemical shifts $\delta_{comb}$ with the weighting factors of Schumann et al. [27] and analyzed the differences $\Delta\delta_{comb}$ between the oxidized and reduced *Pf*Trx-1. The standard deviation to zero $\sigma_0$ was taken as a measure of the significance of the changes. Amino acids with values of $\Delta\delta_{comb}$ above $\sigma_0$ or $2\sigma_0$ and at least partly located on the surface are represented in red and orange in Fig 5 (bottom).

The amino acids probably involved in oxidation-induced conformational changes can also be predicted by $\Delta\delta_{comb}$ values. They are similar but not identical to those observed in the corresponding NMR structures (Fig 5 top). Strongly influenced are again the active center and the subsequent $\alpha_2$ helix (F25 to E41). However, the atoms of the β-bulge do not show significant chemical shift changes after oxidation. In contrast to the direct structural analysis chemical shifts of amino acids between S61 and T74 (located in α-helix $\alpha_3$ and the connecting loop to β-strand $\beta_4$) and between L88 and N91 (loop between helix $\alpha_4$ and β-strand $\beta_5$) reflect additional conformational changes after oxidation. The differences between the conformational changes observed directly from the RMSD and observed from the chemical shift may mainly arise from the fact that the RMSD only used the backbone $C^{\alpha}$-positions but $\Delta\delta_{comb}$ has been calculated for all atoms of a residue including all side chain atoms.

## Interaction sites of *Pf*Trx with other proteins

*Pf*Trx-1 interacts with several proteins that have to recognize the oxidation state of the protein (see e. g. Fig 1). Unfortunately, only one 3D structure of *Pf*Trx with another protein is available, namely the interaction with thioredoxin reductase (*Pf*TrxR) [14]. *Pf*TrxR forms a dimer interacting with 2 molecules of *Pf*TrxR. The interaction sites are depicted in Fig 5 (middle). Functionally, TrxR has to reduce oxidized Trx and thus has to recognize oxidized Trx. Unfortunately, only a crystal structure of an active center mutant of Trx (C30S) is available that simulates the reduced state, that is the TrxR releasing state.

The main interaction site derived from the X-ray structure necessarily is the region around the CGPC motif where the redox reaction should occur. Here, the amino acids E28 to P32 are in direct contact with TrxR. They also show oxidation state-dependent chemical shift and RMSD effects (Fig 5). In addition, amino acids located at the end of α-helix $\alpha_3$ and its connecting loop to β-strand $\beta_4$ (T64, E65, N68 to P73) as well as amino acids of the loop connecting β-strand $\beta_5$ with α-helix $\alpha_4$ (L88 to S93) are in direct contact with thioredoxin reductase. These residues show also significant differences in their $\Delta\delta_{comb}$-values but no larger backbone structural changes as defined by their RMSD values (Fig 5).

## High-pressure response of thioredoxin

The pressure response of *Pf*Trx-1 was studied at 800 MHz based on chemical shift changes in [$^{1}$H-$^{15}$N]-HSQC spectra with pressure. For a study of the effect of the oxidation state on the pressure response two carefully matched samples of completely reduced and completely oxidized *Pf*Trx-1 were produced (see Materials and Methods). For the analysis of the data, in general, two different strategies are applied, a local and a global thermodynamic interpretation. The $^{1}$H and $^{15}$N chemical shifts of the HSQC-cross peaks with pressure observed for *Pf*Trx-1 at 293 K change continuously with pressure as it is typical for fast or intermediate exchange between two or more states. The local interpretation of these chemical shift changes usually describes the pressure dependence by a second-grade polynomial with the first and second-order pressure coefficients $B_1$ and $B_2$. The global interpretation assumes global conformational

states of the whole protein and gives the thermodynamical parameters for the transitions between the states.

For obtaining the specific response of the proteins the pressure coefficients were calculated by correcting the chemical shifts for sequence-specific, pressure-dependent random-coil effects. Since only amide groups were observed, the $^1H$ and $^{15}N$ chemical of a given group was combined. The resulting values $B_1^*$ and $B_2^*$ were plotted as a function of the sequence position and are depicted in Fig 6. Apparently, the oxidation state of the protein has only small effects on the pressure response described by the pressure coefficients.

The average $B_1^*$ coefficients are -0.20 (reduced *Pf*Trx-1) and -0.16 ppm $GPa^{-1}$ (oxidized *Pf*Trx-1), the average $B_2^*$ coefficients are 0.26 (reduced *Pf*Trx-1) and 0.17 ppm $GPa^{-2}$ (oxidized *Pf*Trx-1), that is the average pressure response of the protein is somewhat smaller after oxidation (Fig 6). However, the variations of the individual sequence specific responses are much higher than the differences of the mean values. The corresponding standard deviations of the $B_1^*$ coefficients are 0.42 (reduced *Pf*Trx-1) and 0.45 ppm $GPa^{-1}$ (oxidized *Pf*Trx-1), of the $B_2^*$ coefficients are 0.78 (reduced *Pf*Trx-1) and 0.87 ppm $GPa^{-2}$ (oxidized *Pf*Trx-1).

The ratio of the $B_2/B_1$ values is related to the compressibility factor $\beta'^0 = \beta'(P_0)$ or to the partial molar compressibility $\beta^0 = \beta(P_0)$ and the partial molar volume change $\Delta V^0 = \Delta V(P_0)$ in a two-state system with similar populations ($|\Delta G^0| << 2RT$). It equals $-0.5\Delta\beta'^0/\Delta V^0 = -0.5$ ($<\beta^0>$ + $<V^0>\Delta\beta^0/\Delta V^0$) [33]. Note that the definition of $B_2$ differs by a factor of 2 in the present paper compared to [33]. The average combined $B_2/B_1$ values are -0.48 ppm $GPa^{-1}$ (reduced *Pf*Trx-1) and -0.83 ppm $GPa^{-1}$ (oxidized *Pf*Trx-1), that is the average of $-B_2/B_1$ is significantly larger after oxidation of the protein (Fig 6). These values would give average $\Delta\beta'^0/\Delta V^0$ of 0.96 and 1.66 $GPa^{-1}$, not far away from 1.02 and 1.08 $GPa^{-1}$, the average values of 1 to 2 and the 1 to 3 transitions obtained from the thermodynamical analysis (Table 5). However, the variations of the individual sequence-specific responses are much higher than the differences in the mean values. Unfortunately, the pressure dependence of several residues closes to the active center (grey residues in Fig 7) could not be followed with sufficient accuracy because of line superpositions and pressure-dependent line broadenings. Only at a few points in the 3D structure do the $B_2/B_1$ values change significantly after disulfide bond formation (Fig 7). They are mainly located in the β-strand $\beta_4$ (Fig 7).

## Global conformational equilibria revealed by thermodynamical analysis of the pressure response

A more detailed analysis of the pressure-dependent NMR data can be obtained from a thermodynamical model. Here, a pressure-dependent equilibrium between $N$ global conformational states is assumed. It can be described by pressure-dependent Gibbs free energy differences $\Delta G_{1i}$ (Eq 3). At constant temperature $T$ they can be approximated by the free energy $\Delta G_{1i}^0$ at ($T_0$, $P_0$) and the corresponding differences of the partial molar volumes $\Delta V_{1i}^0$ and the partial molar compressibility factors $\Delta\beta_{1i}'^0$. In addition to these states revealed by their pressure dependence, two different main states $S_r$ and $S_o$ of the reduced and oxidized protein are existing. They are defined by the covalent structure of the protein and are not directly accessible by the analysis of pressure effects but by the NMR structure determination and the analysis of the chemical shifts. In our case, they are considered as pure states and the states derived from the pressure dependence can be considered as sub-states of these main states.

The pressure dependence of the chemical shifts in [$^1H$-$^{15}N$]-HSQC spectra of *Pf*Trx-1 are typical for fast exchange between different conformational states with different chemical shifts $\delta_i$, and can be fitted with Eq 2. An analysis of the chemical shifts clearly indicates that the pressure dependence of some resonance lines cannot be described by the quasi-sigmoidal

dependence characteristic for a two states model. An example is the pressure dependence of the amide [1]H-shifts of E28 in oxidized *Pf*Trx-1 shown in Fig 8. Therefore, we assumed the existence of three conformational states in our analysis. With this assumption, the pressure dependence of all amide [1]H and [15]N shifts could be described satisfactorily.

The line fitting of the individual lines requires the fitting of three parameters ($\delta_1$, $\delta_2$, $\delta_3$) for every line together with six global thermodynamical parameters. An additional difficulty is the limited pressure range experimentally accessible. To obtain meaningful results, this fitting had to be performed iteratively as described before [20, 35]. Here, we had to switch several times between the fit of the local and the global parameters. Finally, all experimental data were fitted perfectly. A typical example is shown in Fig 7. The thermodynamical parameters are represented in Table 5. The complete data sets fitted with these parameters are given in S3 and S4 Files.

At 293 K and 0.1 MPa conformational sub-states $1_{re,ox}$, and $2_{re,ox}$ have quite similar populations since the free energy values are close to the thermal energy R$T$ (2.44 kJ/mol). Oxidation of the cysteine residues leads to an increase of the population of state 2 but this change is not statistically significant. The conformational state 3 has a very low population. At ambient pressure, it is a rare state but gets more populated at high pressure (Table 5). Again, the formation of the disulfide bond has no significant effect on the equilibrium. The functional meaning of the three conformational sub-states is unknown. From their thermodynamical parameters, they cannot distinguished in the reduced and oxidized state of the protein. However, they are not directly related to the disulfide bond formation since we have produced almost pure states by our sample preparation procedure that can also be identified by the chemical shifts of several resonances (Fig 5). Nevertheless, when focusing on individual residues clear differences in the pressure response in the two main states can be observed. An example is E28 (Fig 8) which shows a completely different amide proton response when oxidized. It is located close to W27 and also shows typical chemical shift and structural changes after oxidation of Trx.

What structural changes are coupled to the transition from sub-state 1 to sub-state 2 or 3? The optimal answer would be obtained from the determination of their three-dimensional structures by high-pressure NMR or possibly by high-pressure crystallography [36]. However, high-pressure crystallography has the disadvantage that it is difficult to correlate the transitions in the crystal with those in solution (see e. g. Girard et al. [37]). A qualitative way to associate possible structural changes with the different transitions is the use of pressure-induced chemical shift changes. Here, the basic assumption is that there exist positive correlations between larger chemical shift changes with larger structural changes. Since the fitting of the data gives also the "pure" chemical shifts ($\delta_1$, $\delta_2$, $\delta_3$) in a given state, the corresponding shift changes can be used to categorize the different transitions structurally (Fig 9). If for a given residue $|\delta_2 - \delta_1|$ $\geq 2\,|\delta_3 - \delta_2|$ the induced structural change is suggested to be due to the transition to 2 (depicted in red Fig 9). When $|\delta_3 - \delta_2| > 2\,|\delta_2 - \delta_1|$ it is assigned to the rare state 3. The transition dominant at normal pressure encompasses most of the residues of the protein in the reduced and oxidized form with a focus on the α-helices $\alpha_1$, $\alpha_2$, and $\alpha_4$, to the β-strands $\beta_2$ and $\beta_3$ and the redox center. The rare transition is dominantly found only for 12 to 13% of the residues with a main focus on the β-strand $\beta_5$ and the loop connecting it to α-helix $\alpha_4$ (T86 to N91). This stretch of amino acids is also involved in the interaction with Trx-reductase and show also significant chemical shift changes with disulfide bond formation (Fig 5). Independent of their structural interpretation, almost the same residues are selected for the reduced and oxidized protein, indicating again that the conformational equilibria detected by the pressure perturbation have similar functions in the reduced and oxidized thioredoxin. A plausible explanation for the conformational sub-states detected by NMR is that they represent the structural adaptions that are necessary for the binding of interacting proteins such as the *Pf*Trx-reductase.

## Conclusions

*Pf*Trx-1 shows very well-resolved homo- and heteronuclear NMR spectra giving rise to well-resolved three-dimensional structures of the reduced and oxidized protein in solution. The general fold is conserved in the two redox states $S_r$ and $S_o$. The structural differences between the two main states defined by the $C^\alpha$-RMSD mainly concern the active center loop and large parts of α-helix $\alpha_2$ (E28 to E42). In addition, regions distant from the active center (Figs 3 and 4) show significant structural changes after oxidation (N-terminal end of $\alpha_3$ and $\beta_5$). These structural differences are also supported by a chemical shift analysis encompassing all atoms (main and side chain) of the protein.

The thermodynamical analysis of the high-pressure data reveals the existence of 3 sub-states of $S_r$ and $S_o$ in both forms of the protein. They could be required to allow the structural adaption of the surface of the protein for optimal binding of interacting proteins. Such an effect has been described earlier for the Ras-protein [38].

## Supporting information

**S1 File. Validation report for the structure of the reduced thioredoxin-1.**
(PDF)

**S2 File. Validation report for the structure of the oxidized thioredoxin-1.**
(PDF)

**S3 File. Pressure induced chemical shift changes of reduced thioredoxin-1.**
(PDF)

**S4 File. Pressure induced chemical shift changes of oxidized thioredoxin-1.**
(PDF)

## Acknowledgments

We gratefully acknowledge the valuable contributions of R. Heiner Schirmer and Katja Becker who started this work with us. Unfortunately, R. H. Schirmer deceased too early to see the completion of the project. We thank Daniela Krämer for her contributions in data evaluation.

## Author Contributions

**Conceptualization:** Hans Robert Kalbitzer.

**Formal analysis:** Hans Robert Kalbitzer.

**Funding acquisition:** Hans Robert Kalbitzer.

**Investigation:** Claudia Elisabeth Munte.

**Methodology:** Hans Robert Kalbitzer.

**Project administration:** Claudia Elisabeth Munte, Hans Robert Kalbitzer.

**Software:** Claudia Elisabeth Munte.

**Supervision:** Hans Robert Kalbitzer.

**Validation:** Hans Robert Kalbitzer.

**Writing – original draft:** Claudia Elisabeth Munte, Hans Robert Kalbitzer.

**Writing – review & editing:** Claudia Elisabeth Munte, Hans Robert Kalbitzer.

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
