## [Decision Letter · Decision Letter 0]

9 Feb 2024

PONE-D-23-42130Solution Structure and Pressure Response of Thioredoxin-1 of Plasmodium falciparumPLOS ONE

Dear Dr. Kalbitzer,

Thank you for submitting your manuscript to PLOS ONE. After careful consideration, we feel that it has merit but does not fully meet PLOS ONE’s publication criteria as it currently stands. Therefore, we invite you to submit a revised version of the manuscript that addresses the points raised during the review process.

We look forward to receiving your revised manuscript.

Kind regards,

Ravi Pratap Barnwal, Ph.D.

Academic Editor

PLOS ONE

Journal Requirements:

2. We note that this submission includes NMR spectroscopy data. We would recommend that you include the following information in your methods section or as Supporting Information files:

1) The make/source of the NMR instrument used in your study, as well as the magnetic field strength. For each individual experiment, please also list: the nucleus being measured; the sample concentration; the solvent in which the sample is dissolved and if solvent signal suppression was used; the reference standard and the temperature.

2) A list of the chemical shifts for all compounds characterised by NMR spectroscopy, specifying, where relevant: the chemical shift (δ), the multiplicity and the coupling constants (in Hz), for the appropriate nuclei used for assignment.

3)The full integrated NMR spectrum, clearly labelled with the compound name and chemical structure.

We also strongly encourage authors to provide primary NMR data files, in particular for new compounds which have not been characterised in the existing literature. Authors should provide the acquisition data, FID files and processing parameters for each experiment, clearly labelled with the compound name and identifier, as well as a structure file for each provided dataset. See our list of recommended repositories here: https://journals.plos.org/plosone/s/recommended-repositories.

"DFG (KA647/22-1) Funding of HRK"

5. Please expand the acronym “DFG” (as indicated in your financial disclosure) so that it states the name of your funders in full.

"This work has been supported by the DFG (KA647/22-1). We gratefully acknowledge the valuable contributions of R. Heiner Schirmer and Katja Becker who started this work with us. Unfortunately, R. H. Schirmer deceased too early to see the completion of the project. We thank Daniela Krämer for her help with the data evaluation."

"DFG (KA647/22-1) Funding of HRK"

Reviewers' comments:

Reviewer's Responses to Questions

**Comments to the Author**

1. Is the manuscript technically sound, and do the data support the conclusions?

Reviewer #1: Yes

2. Has the statistical analysis been performed appropriately and rigorously? 

Reviewer #1: Yes

3. Have the authors made all data underlying the findings in their manuscript fully available?

Reviewer #1: Yes

4. Is the manuscript presented in an intelligible fashion and written in standard English?

Reviewer #1: Yes

5. Review Comments to the Author

Reviewer #1: In this manuscript by Munte et al., the authors described the structural characterization and pressure-induced response of Thioredoxin-1 (reduced and oxidized state) in Plasmodium falciparum using solution NMR techniques. Both the reduced and oxidized states of Thioredoxin-1 showed a common structural fold with minor structural differences. They evaluated the pressure-induced response of Thioredoxin-1 in its reduced and oxidized states in a pressure range of 0.1 MPa to 200 MPa. This manuscript is well written. I recommend this article for the publication in “PLOS ONE” with minor revisions. The following corrections have been suggested:

1. There are some typographical errors in the manuscript and should be corrected.

2. Keywords should be in alphabetical order.

3. Sentence “These three states are sub-states from the two main states of the protein, the reduced and oxidized states Sr and So” (line 42-43) should be modified.

4. The typo error in the sentence “Because of the required additivity of the chemical shifts the Euclidean distance has been used in the]se calculations.” should be corrected (line 194).

5. Sentence “The Ramachandran Plot resulted in additional allowed regions, 0.9% in generously allowed regions, and 0.0% in disallowed regions” should be modified to include information on most favorable regions (line 313-315).

6. PLOS authors have the option to publish the peer review history of their article (what does this mean?). If published, this will include your full peer review and any attached files.

Reviewer #1: No

---

## [Author Response · Author response to Decision Letter 0]

5 Mar 2024

Dear Dr. Ravi Pratap Barnwal,

We have corrected the manuscript according to your and the reviewer’s suggestions. We added all necessary experimental conditions. Especially, a complete chemical shift list has already been published and has been stored in the BioMagResBank, accession numbers 6442 and 16147. We also corrected the funding information and removed it from Acknowledgements. Please add to the funding information the sentence "The funders had no role in study design, data collection and analysis, decision to publish, or preparation of the manuscript". We also gave the full name of DFG, namely Deutsche Forschungsgemeinschaft.

Reviewer 1

1. There are some typographical errors in the manuscript and should be corrected.

 We screened the text for typographical errors.

 2. Keywords should be in alphabetical order. 

 We ordered them as suggested.

3. Sentence “These three states are sub-states from the two main states of the protein, the reduced and oxidized states Sr and So” (line 42-43) should be modified. 

 We modified the sentence and replaced it by “These three conformational states were identified in the reduced as well as in the oxidized form of the protein, therefore, they represent sub-states of the two main oxidation states of PfTrx-1.” We hope the meaning is now clearer 

4. The typo error in the sentence “Because of the required additivity of the chemical shifts the Euclidean distance has been used in the]se calculations.” should be corrected (line 194).

We corrected this.

 5. Sentence “The Ramachandran Plot resulted in additional allowed regions, 0.9% in generously allowed regions, and 0.0% in disallowed regions” should be modified to include information on most favorable regions (line 313-315).

We added this information.

---

## [Decision Letter · Decision Letter 1]

20 Mar 2024

Solution Structure and Pressure Response of Thioredoxin-1 of Plasmodium falciparum

PONE-D-23-42130R1

Dear Dr. Kalbitzer,

We’re pleased to inform you that your manuscript has been judged scientifically suitable for publication and will be formally accepted for publication once it meets all outstanding technical requirements.

An invoice for payment will follow shortly after the formal acceptance. To ensure an efficient process, please log into Editorial Manager at Editorial Manager® , click the 'Update My Information' link at the top of the page, and double check that your user information is up-to-date. If you have any billing related questions, please contact our Author Billing department directly at authorbilling@plos.org.

Kind regards,

Ravi Pratap Barnwal, Ph.D.

Academic Editor

PLOS ONE

Additional Editor Comments (optional):

Reviewers' comments:

Reviewer's Responses to Questions

**Comments to the Author**

1. If the authors have adequately addressed your comments raised in a previous round of review and you feel that this manuscript is now acceptable for publication, you may indicate that here to bypass the “Comments to the Author” section, enter your conflict of interest statement in the “Confidential to Editor” section, and submit your "Accept" recommendation.

Reviewer #1: All comments have been addressed

2. Is the manuscript technically sound, and do the data support the conclusions?

Reviewer #1: Yes

3. Has the statistical analysis been performed appropriately and rigorously? 

Reviewer #1: Yes

4. Have the authors made all data underlying the findings in their manuscript fully available?

Reviewer #1: Yes

5. Is the manuscript presented in an intelligible fashion and written in standard English?

Reviewer #1: Yes

6. Review Comments to the Author

Reviewer #1: (No Response)

7. PLOS authors have the option to publish the peer review history of their article (what does this mean?). If published, this will include your full peer review and any attached files.

Reviewer #1: **Yes: **Dr. Janeka Gartia

---

## [Editor Report · Acceptance letter]

28 Mar 2024

PONE-D-23-42130R1 

PLOS ONE

Dear Dr. Kalbitzer, 

I'm pleased to inform you that your manuscript has been deemed suitable for publication in PLOS ONE. Congratulations! Your manuscript is now being handed over to our production team.

Kind regards, 

on behalf of

Dr. Ravi Pratap Barnwal 

Academic Editor

PLOS ONE